# *mixup*: Beyond Empirical Risk Minimization

**Hongyi Zhang**
MIT

**Moustapha Cisse, Yann N. Dauphin, David Lopez-Paz**[*]
FAIR

## Abstract

Large deep neural networks are powerful, but exhibit undesirable behaviors such as memorization and sensitivity to adversarial examples. In this work, we propose *mixup*, a simple learning principle to alleviate these issues. In essence, *mixup* trains a neural network on convex combinations of pairs of examples and their labels. By doing so, *mixup* regularizes the neural network to favor simple linear behavior in-between training examples. Our experiments on the ImageNet-2012, CIFAR-10, CIFAR-100, Google commands and UCI datasets show that *mixup* improves the generalization of state-of-the-art neural network architectures. We also find that *mixup* reduces the memorization of corrupt labels, increases the robustness to adversarial examples, and stabilizes the training of generative adversarial networks.

## 1 Introduction

Large deep neural networks have enabled breakthroughs in fields such as computer vision (Krizhevsky et al., 2012), speech recognition (Hinton et al., 2012), and reinforcement learning (Silver et al., 2016). In most successful applications, these neural networks share two commonalities. First, they are trained as to minimize their average error over the training data, a learning rule also known as the Empirical Risk Minimization (ERM) principle (Vapnik, 1998). Second, the size of these state-of-the-art neural networks scales linearly with the number of training examples. For instance, the network of Springenberg et al. (2015) used $10^6$ parameters to model the $5 \cdot 10^4$ images in the CIFAR-10 dataset, the network of (Simonyan & Zisserman, 2015) used $10^8$ parameters to model the $10^6$ images in the ImageNet-2012 dataset, and the network of Chelba et al. (2013) used $2 \cdot 10^{10}$ parameters to model the $10^9$ words in the One Billion Word dataset.

Strikingly, a classical result in learning theory (Vapnik & Chervonenkis, 1971) tells us that the convergence of ERM is guaranteed as long as the size of the learning machine (e.g., the neural network) does not increase with the number of training data. Here, the size of a learning machine is measured in terms of its number of parameters or, relatedly, its VC-complexity (Harvey et al., 2017).

This contradiction challenges the suitability of ERM to train our current neural network models, as highlighted in recent research. On the one hand, ERM allows large neural networks to *memorize* (instead of *generalize* from) the training data even in the presence of strong regularization, or in classification problems where the labels are assigned at random (Zhang et al., 2017). On the other hand, neural networks trained with ERM change their predictions drastically when evaluated on examples just outside the training distribution (Szegedy et al., 2014), also known as *adversarial examples*. This evidence suggests that ERM is unable to explain or provide generalization on testing distributions that differ *only slightly* from the training data. However, what is the alternative to ERM?

The method of choice to train on similar but different examples to the training data is known as *data augmentation* (Simard et al., 1998), formalized by the Vicinal Risk Minimization (VRM) principle (Chapelle et al., 2000). In VRM, human knowledge is required to describe a *vicinity* or neighborhood around each example in the training data. Then, additional *virtual* examples can be drawn from the vicinity distribution of the training examples to enlarge the support of the training distribution. For instance, when performing image classification, it is common to define the vicinity of one image as the set of its horizontal reflections, slight rotations, and mild scalings. While data augmentation consistently leads to improved generalization (Simard et al., 1998), the procedure is dataset-dependent, and thus requires the use of expert knowledge. Furthermore, data augmentation assumes that the

---

[*]Alphabetical order.

examples in the vicinity share the same class, and does not model the vicinity relation across examples of different classes.

**Contribution**     Motivated by these issues, we introduce a simple and data-agnostic data augmentation routine, termed *mixup* (Section 2). In a nutshell, *mixup* constructs virtual training examples

$$\tilde{x} = \lambda x_i + (1 - \lambda) x_j, \qquad \text{where } x_i, x_j \text{ are raw input vectors}$$
$$\tilde{y} = \lambda y_i + (1 - \lambda) y_j, \qquad \text{where } y_i, y_j \text{ are one-hot label encodings}$$

$(x_i, y_i)$ and $(x_j, y_j)$ are two examples drawn at random from our training data, and $\lambda \in [0, 1]$. Therefore, *mixup* extends the training distribution by incorporating the prior knowledge that linear interpolations of feature vectors should lead to linear interpolations of the associated targets. *mixup* can be implemented in a few lines of code, and introduces minimal computation overhead.

Despite its simplicity, *mixup* allows a new state-of-the-art performance in the CIFAR-10, CIFAR-100, and ImageNet-2012 image classification datasets (Sections 3.1 and 3.2). Furthermore, *mixup* increases the robustness of neural networks when learning from corrupt labels (Section 3.4), or facing adversarial examples (Section 3.5). Finally, *mixup* improves generalization on speech (Sections 3.3) and tabular (Section 3.6) data, and can be used to stabilize the training of GANs (Section 3.7). The source-code necessary to replicate our CIFAR-10 experiments is available at:

```
https://github.com/facebookresearch/mixup-cifar10.
```

To understand the effects of various design choices in *mixup*, we conduct a thorough set of ablation study experiments (Section 3.8). The results suggest that *mixup* performs significantly better than related methods in previous work, and each of the design choices contributes to the final performance. We conclude by exploring the connections to prior work (Section 4), as well as offering some points for discussion (Section 5).

## 2     FROM EMPIRICAL RISK MINIMIZATION TO *mixup*

In supervised learning, we are interested in finding a function $f \in \mathcal{F}$ that describes the relationship between a random feature vector $X$ and a random target vector $Y$, which follow the joint distribution $P(X, Y)$. To this end, we first define a loss function $\ell$ that penalizes the differences between predictions $f(x)$ and actual targets $y$, for examples $(x, y) \sim P$. Then, we minimize the average of the loss function $\ell$ over the data distribution $P$, also known as the *expected risk*:

$$R(f) = \int \ell(f(x), y) \mathrm{d}P(x, y).$$

Unfortunately, the distribution $P$ is unknown in most practical situations. Instead, we usually have access to a set of training data $\mathcal{D} = \{(x_i, y_i)\}_{i=1}^n$, where $(x_i, y_i) \sim P$ for all $i = 1, \ldots, n$. Using the training data $\mathcal{D}$, we may approximate $P$ by the *empirical distribution*

$$P_\delta(x, y) = \frac{1}{n} \sum_{i=1}^n \delta(x = x_i, y = y_i),$$

where $\delta(x = x_i, y = y_i)$ is a Dirac mass centered at $(x_i, y_i)$. Using the empirical distribution $P_\delta$, we can now approximate the expected risk by the *empirical risk*:

$$R_\delta(f) = \int \ell(f(x), y) \mathrm{d}P_\delta(x, y) = \frac{1}{n} \sum_{i=1}^n \ell(f(x_i), y_i). \tag{1}$$

Learning the function $f$ by minimizing (1) is known as the Empirical Risk Minimization (ERM) principle (Vapnik, 1998). While efficient to compute, the empirical risk (1) monitors the behaviour of $f$ only at a finite set of $n$ examples. When considering functions with a number parameters comparable to $n$ (such as large neural networks), one trivial way to minimize (1) is to memorize the training data (Zhang et al., 2017). Memorization, in turn, leads to the undesirable behaviour of $f$ outside the training data (Szegedy et al., 2014).

```
# y1, y2 should be one-hot vectors
for (x1, y1), (x2, y2) in zip(loader1, loader2):
    lam = numpy.random.beta(alpha, alpha)
    x = Variable(lam * x1 + (1. - lam) * x2)
    y = Variable(lam * y1 + (1. - lam) * y2)
    optimizer.zero_grad()
    loss(net(x), y).backward()
    optimizer.step()
```

(a) One epoch of *mixup* training in PyTorch.

(b) Effect of *mixup* ($\alpha = 1$) on a toy problem. Green: Class 0. Orange: Class 1. Blue shading indicates $p(y = 1|x)$.

Figure 1: Illustration of *mixup*, which converges to ERM as $\alpha \to 0$.

However, the naïve estimate $P_\delta$ is one out of many possible choices to approximate the true distribution $P$. For instance, in the *Vicinal Risk Minimization* (VRM) principle (Chapelle et al., 2000), the distribution $P$ is approximated by

$$P_\nu(\tilde{x}, \tilde{y}) = \frac{1}{n} \sum_{i=1}^{n} \nu(\tilde{x}, \tilde{y}|x_i, y_i),$$

where $\nu$ is a *vicinity distribution* that measures the probability of finding the *virtual* feature-target pair $(\tilde{x}, \tilde{y})$ in the *vicinity* of the training feature-target pair $(x_i, y_i)$. In particular, Chapelle et al. (2000) considered Gaussian vicinities $\nu(\tilde{x}, \tilde{y}|x_i, y_i) = \mathcal{N}(\tilde{x} - x_i, \sigma^2)\delta(\tilde{y} = y_i)$, which is equivalent to augmenting the training data with additive Gaussian noise. To learn using VRM, we sample the vicinal distribution to construct a dataset $\mathcal{D}_\nu := \{(\tilde{x}_i, \tilde{y}_i)\}_{i=1}^{m}$, and minimize the *empirical vicinal risk*:

$$R_\nu(f) = \frac{1}{m} \sum_{i=1}^{m} \ell(f(\tilde{x}_i), \tilde{y}_i).$$

The contribution of this paper is to propose a generic vicinal distribution, called *mixup*:

$$\mu(\tilde{x}, \tilde{y}|x_i, y_i) = \frac{1}{n} \sum_{j}^{n} \mathbb{E}_\lambda \left[ \delta(\tilde{x} = \lambda \cdot x_i + (1 - \lambda) \cdot x_j, \tilde{y} = \lambda \cdot y_i + (1 - \lambda) \cdot y_j) \right],$$

where $\lambda \sim \text{Beta}(\alpha, \alpha)$, for $\alpha \in (0, \infty)$. In a nutshell, sampling from the *mixup* vicinal distribution produces virtual feature-target vectors

$$\tilde{x} = \lambda x_i + (1 - \lambda)x_j,$$
$$\tilde{y} = \lambda y_i + (1 - \lambda)y_j,$$

where $(x_i, y_i)$ and $(x_j, y_j)$ are two feature-target vectors drawn at random from the training data, and $\lambda \in [0, 1]$. The *mixup* hyper-parameter $\alpha$ controls the strength of interpolation between feature-target pairs, recovering the ERM principle as $\alpha \to 0$.

The implementation of *mixup* training is straightforward, and introduces a minimal computation overhead. Figure 1a shows the few lines of code necessary to implement *mixup* training in PyTorch. Finally, we mention alternative design choices. First, in preliminary experiments we find that convex combinations of three or more examples with weights sampled from a Dirichlet distribution does not provide further gain, but increases the computation cost of *mixup*. Second, our current implementation uses a single data loader to obtain one minibatch, and then *mixup* is applied to the same minibatch after random shuffling. We found this strategy works equally well, while reducing I/O requirements. Third, interpolating only between inputs with equal label did not lead to the performance gains of *mixup* discussed in the sequel. More empirical comparison can be found in Section 3.8.

**What is *mixup* doing?** The *mixup* vicinal distribution can be understood as a form of data augmentation that encourages the model $f$ to behave linearly in-between training examples. We argue that this linear behaviour reduces the amount of undesirable oscillations when predicting outside the training examples. Also, linearity is a good inductive bias from the perspective of Occam's razor,

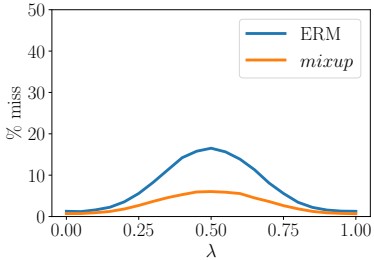 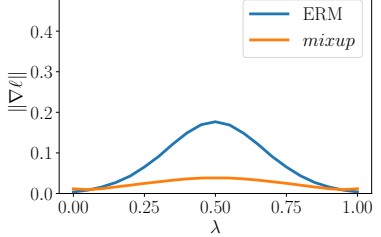

(a) Prediction errors in-between training data. Evaluated at $x = \lambda x_i + (1-\lambda)x_j$, a prediction is counted as a "miss" if it does not belong to $\{y_i, y_j\}$. The model trained with *mixup* has fewer misses.

(b) Norm of the gradients of the model w.r.t. input in-between training data, evaluated at $x = \lambda x_i + (1-\lambda)x_j$. The model trained with *mixup* has smaller gradient norms.

Figure 2: *mixup* leads to more robust model behaviors in-between the training data.

| Model | Method | Epochs | Top-1 Error | Top-5 Error |
|---|---|---|---|---|
| ResNet-50 | ERM (Goyal et al., 2017) | 90 | 23.5 | - |
| | *mixup* $\alpha = 0.2$ | 90 | **23.3** | **6.6** |
| ResNet-101 | ERM (Goyal et al., 2017) | 90 | 22.1 | - |
| | *mixup* $\alpha = 0.2$ | 90 | **21.5** | **5.6** |
| ResNeXt-101 32*4d | ERM (Xie et al., 2016) | 100 | 21.2 | - |
| | ERM | 90 | 21.2 | 5.6 |
| | *mixup* $\alpha = 0.4$ | 90 | **20.7** | **5.3** |
| ResNeXt-101 64*4d | ERM (Xie et al., 2016) | 100 | 20.4 | 5.3 |
| | *mixup* $\alpha = 0.4$ | 90 | **19.8** | **4.9** |
| ResNet-50 | ERM | 200 | 23.6 | 7.0 |
| | *mixup* $\alpha = 0.2$ | 200 | **22.1** | **6.1** |
| ResNet-101 | ERM | 200 | 22.0 | 6.1 |
| | *mixup* $\alpha = 0.2$ | 200 | **20.8** | **5.4** |
| ResNeXt-101 32*4d | ERM | 200 | 21.3 | 5.9 |
| | *mixup* $\alpha = 0.4$ | 200 | **20.1** | **5.0** |

Table 1: Validation errors for ERM and *mixup* on the development set of ImageNet-2012.

since it is one of the simplest possible behaviors. Figure 1b shows that *mixup* leads to decision boundaries that transition linearly from class to class, providing a smoother estimate of uncertainty. Figure 2 illustrate the average behaviors of two neural network models trained on the CIFAR-10 dataset using ERM and *mixup*. Both models have the same architecture, are trained with the same procedure, and are evaluated at the same points in-between randomly sampled training data. The model trained with *mixup* is more stable in terms of model predictions and gradient norms in-between training samples.

## 3 EXPERIMENTS

### 3.1 IMAGENET CLASSIFICATION

We evaluate *mixup* on the ImageNet-2012 classification dataset (Russakovsky et al., 2015). This dataset contains 1.3 million training images and 50,000 validation images, from a total of 1,000 classes. For training, we follow standard data augmentation practices: scale and aspect ratio distortions, random crops, and horizontal flips (Goyal et al., 2017). During evaluation, only the $224 \times 224$ central crop of each image is tested. We use *mixup* and ERM to train several state-of-the-art ImageNet-2012 classification models, and report both top-1 and top-5 error rates in Table 1.

| Dataset | Model | ERM | *mixup* |
|---------|-------|-----|---------|
| CIFAR-10 | PreAct ResNet-18 | 5.6 | **4.2** |
| | WideResNet-28-10 | 3.8 | **2.7** |
| | DenseNet-BC-190 | 3.7 | **2.7** |
| CIFAR-100 | PreAct ResNet-18 | 25.6 | **21.1** |
| | WideResNet-28-10 | 19.4 | **17.5** |
| | DenseNet-BC-190 | 19.0 | **16.8** |

(a) Test errors for the CIFAR experiments.

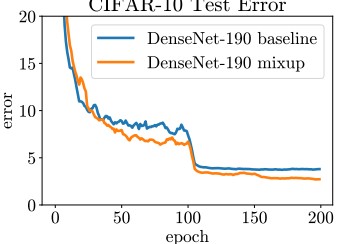

(b) Test error evolution for the best ERM and *mixup* models.

Figure 3: Test errors for ERM and *mixup* on the CIFAR experiments.

For all the experiments in this section, we use data-parallel distributed training in Caffe2[1] with a minibatch size of 1,024. We use the learning rate schedule described in (Goyal et al., 2017). Specifically, the learning rate is increased linearly from 0.1 to 0.4 during the first 5 epochs, and it is then divided by 10 after 30, 60 and 80 epochs when training for 90 epochs; or after 60, 120 and 180 epochs when training for 200 epochs.

For *mixup*, we find that $\alpha \in [0.1, 0.4]$ leads to improved performance over ERM, whereas for large $\alpha$, *mixup* leads to underfitting. We also find that models with higher capacities and/or longer training runs are the ones to benefit the most from *mixup*. For example, when trained for 90 epochs, the *mixup* variants of ResNet-101 and ResNeXt-101 obtain a greater improvement (0.5% to 0.6%) over their ERM analogues than the gain of smaller models such as ResNet-50 (0.2%). When trained for 200 epochs, the top-1 error of the *mixup* variant of ResNet-50 is further reduced by 1.2% compared to the 90 epoch run, whereas its ERM analogue stays the same.

## 3.2 CIFAR-10 AND CIFAR-100

We conduct additional image classification experiments on the CIFAR-10 and CIFAR-100 datasets to further evaluate the generalization performance of *mixup*. In particular, we compare ERM and *mixup* training for: PreAct ResNet-18 (He et al., 2016) as implemented in (Liu, 2017), WideResNet-28-10 (Zagoruyko & Komodakis, 2016a) as implemented in (Zagoruyko & Komodakis, 2016b), and DenseNet (Huang et al., 2017) as implemented in (Veit, 2017). For DenseNet, we change the growth rate to 40 to follow the DenseNet-BC-190 specification from (Huang et al., 2017). For *mixup*, we fix $\alpha = 1$, which results in interpolations $\lambda$ uniformly distributed between zero and one. All models are trained on a single Nvidia Tesla P100 GPU using PyTorch[2] for 200 epochs on the training set with 128 examples per minibatch, and evaluated on the test set. Learning rates start at 0.1 and are divided by 10 after 100 and 150 epochs for all models except WideResNet. For WideResNet, we follow (Zagoruyko & Komodakis, 2016a) and divide the learning rate by 10 after 60, 120 and 180 epochs. Weight decay is set to $10^{-4}$. We do not use dropout in these experiments.

We summarize our results in Figure 3a. In both CIFAR-10 and CIFAR-100 classification problems, the models trained using *mixup* significantly outperform their analogues trained with ERM. As seen in Figure 3b, *mixup* and ERM converge at a similar speed to their best test errors. Note that the DenseNet models in (Huang et al., 2017) were trained for 300 epochs with further learning rate decays scheduled at the 150 and 225 epochs, which may explain the discrepancy the performance of DenseNet reported in Figure 3a and the original result of Huang et al. (2017).

## 3.3 SPEECH DATA

Next, we perform speech recognition experiments using the Google commands dataset (Warden, 2017). The dataset contains 65,000 utterances, where each utterance is about one-second long and belongs to one out of 30 classes. The classes correspond to voice commands such as *yes, no, down, left*, as pronounced by a few thousand different speakers. To preprocess the utterances, we first

---

[1] https://caffe2.ai
[2] http://pytorch.org

| Model | Method | Validation set | Test set |
|---|---|---|---|
| LeNet | ERM | **9.8** | **10.3** |
| | *mixup* ($\alpha = 0.1$) | 10.1 | 10.8 |
| | *mixup* ($\alpha = 0.2$) | 10.2 | 11.3 |
| VGG-11 | ERM | 5.0 | 4.6 |
| | *mixup* ($\alpha = 0.1$) | 4.0 | 3.8 |
| | *mixup* ($\alpha = 0.2$) | **3.9** | **3.4** |

Figure 4: Classification errors of ERM and *mixup* on the Google commands dataset.

extract normalized spectrograms from the original waveforms at a sampling rate of 16 kHz. Next, we zero-pad the spectrograms to equalize their sizes at $160 \times 101$. For speech data, it is reasonable to apply *mixup* both at the waveform and spectrogram levels. Here, we apply *mixup* at the spectrogram level just before feeding the data to the network.

For this experiment, we compare a LeNet (Lecun et al., 2001) and a VGG-11 (Simonyan & Zisserman, 2015) architecture, each of them composed by two convolutional and two fully-connected layers. We train each model for 30 epochs with minibatches of 100 examples, using Adam as the optimizer (Kingma & Ba, 2015). Training starts with a learning rate equal to $3 \times 10^{-3}$ and is divided by 10 every 10 epochs. For *mixup*, we use a warm-up period of five epochs where we train the network on original training examples, since we find it speeds up initial convergence. Table 4 shows that *mixup* outperforms ERM on this task, specially when using VGG-11, the model with larger capacity.

## 3.4 MEMORIZATION OF CORRUPTED LABELS

Following Zhang et al. (2017), we evaluate the robustness of ERM and *mixup* models against randomly corrupted labels. We hypothesize that increasing the strength of *mixup* interpolation $\alpha$ should generate virtual examples further from the training examples, making memorization more difficult to achieve. In particular, it should be easier to learn interpolations between real examples compared to memorizing interpolations involving random labels. We adapt an open-source implementation (Zhang, 2017) to generate three CIFAR-10 training sets, where 20%, 50%, or 80% of the labels are replaced by random noise, respectively. All the test labels are kept intact for evaluation. Dropout (Srivastava et al., 2014) is considered the state-of-the-art method for learning with corrupted labels (Arpit et al., 2017). Thus, we compare in these experiments *mixup*, dropout, *mixup* + dropout, and ERM. For *mixup*, we choose $\alpha \in \{1, 2, 8, 32\}$; for dropout, we add one dropout layer in each PreAct block after the ReLU activation layer between two convolution layers, as suggested in (Zagoruyko & Komodakis, 2016a). We choose the dropout probability $p \in \{0.5, 0.7, 0.8, 0.9\}$. For the combination of *mixup* and dropout, we choose $\alpha \in \{1, 2, 4, 8\}$ and $p \in \{0.3, 0.5, 0.7\}$. These experiments use the PreAct ResNet-18 (He et al., 2016) model implemented in (Liu, 2017). All the other settings are the same as in Section 3.2.

We summarize our results in Table 2, where we note the best test error achieved during the training session, as well as the final test error after 200 epochs. To quantify the amount of memorization, we also evaluate the training errors at the last epoch on real labels and corrupted labels. As the training progresses with a smaller learning rate (e.g. less than 0.01), the ERM model starts to overfit the corrupted labels. When using a large probability (e.g. 0.7 or 0.8), dropout can effectively reduce overfitting. *mixup* with a large $\alpha$ (e.g. 8 or 32) outperforms dropout on both the best and last epoch test errors, and achieves lower training error on real labels while remaining resistant to noisy labels. Interestingly, *mixup* + dropout performs the best of all, showing that the two methods are compatible.

## 3.5 ROBUSTNESS TO ADVERSARIAL EXAMPLES

One undesirable consequence of models trained using ERM is their fragility to adversarial examples (Szegedy et al., 2014). Adversarial examples are obtained by adding tiny (visually imperceptible) perturbations to legitimate examples in order to deteriorate the performance of the model. The adversarial noise is generated by ascending the gradient of the loss surface with respect to the legitimate example. Improving the robustness to adversarial examples is a topic of active research.

| Label corruption | Method | Test error | | Training error | |
|---|---|---|---|---|---|
| | | Best | Last | Real | Corrupted |
| 20% | ERM | 12.7 | 16.6 | 0.05 | 0.28 |
| | ERM + dropout ($p = 0.7$) | 8.8 | 10.4 | 5.26 | 83.55 |
| | *mixup* ($\alpha = 8$) | **5.9** | 6.4 | 2.27 | 86.32 |
| | *mixup* + dropout ($\alpha = 4, p = 0.1$) | 6.2 | **6.2** | 1.92 | 85.02 |
| 50% | ERM | 18.8 | 44.6 | 0.26 | 0.64 |
| | ERM + dropout ($p = 0.8$) | 14.1 | 15.5 | 12.71 | 86.98 |
| | *mixup* ($\alpha = 32$) | 11.3 | 12.7 | 5.84 | 85.71 |
| | *mixup* + dropout ($\alpha = 8, p = 0.3$) | **10.9** | **10.9** | 7.56 | 87.90 |
| 80% | ERM | 36.5 | 73.9 | 0.62 | 0.83 |
| | ERM + dropout ($p = 0.8$) | 30.9 | 35.1 | 29.84 | 86.37 |
| | *mixup* ($\alpha = 32$) | 25.3 | 30.9 | 18.92 | 85.44 |
| | *mixup* + dropout ($\alpha = 8, p = 0.3$) | **24.0** | **24.8** | 19.70 | 87.67 |

Table 2: Results on the corrupted label experiments for the best models.

| Metric | Method | FGSM | I-FGSM |
|---|---|---|---|
| Top-1 | ERM | 90.7 | 99.9 |
| | *mixup* | **75.2** | 99.6 |
| Top-5 | ERM | 63.1 | 93.4 |
| | *mixup* | **49.1** | 95.8 |

(a) White box attacks.

| Metric | Method | FGSM | I-FGSM |
|---|---|---|---|
| Top-1 | ERM | 57.0 | 57.3 |
| | *mixup* | **46.0** | **40.9** |
| Top-5 | ERM | 24.8 | 18.1 |
| | *mixup* | **17.4** | **11.8** |

(b) Black box attacks.

Table 3: Classification errors of ERM and *mixup* models when tested on adversarial examples.

Among the several methods aiming to solve this problem, some have proposed to penalize the norm of the Jacobian of the model to control its Lipschitz constant (Drucker & Le Cun, 1992; Cisse et al., 2017; Bartlett et al., 2017; Hein & Andriushchenko, 2017). Other approaches perform data augmentation by producing and training on adversarial examples (Goodfellow et al., 2015). Unfortunately, all of these methods add significant computational overhead to ERM. Here, we show that *mixup* can significantly improve the robustness of neural networks without hindering the speed of ERM by penalizing the norm of the gradient of the loss w.r.t a given input along the most plausible directions (e.g. the directions to other training points). Indeed, Figure 2 shows that *mixup* results in models having a smaller loss and gradient norm between examples compared to vanilla ERM.

To assess the robustness of *mixup* models to adversarial examples, we use three ResNet-101 models: two of them trained using ERM on ImageNet-2012, and the third trained using *mixup*. In the first set of experiments, we study the robustness of one ERM model and the *mixup* model against white box attacks. That is, for each of the two models, we use the model itself to generate adversarial examples, either using the Fast Gradient Sign Method (FGSM) or the Iterative FGSM (I-FGSM) methods (Goodfellow et al., 2015), allowing a maximum perturbation of $\epsilon = 4$ for every pixel. For I-FGSM, we use 10 iterations with equal step size. In the second set of experiments, we evaluate robustness against black box attacks. That is, we use the first ERM model to produce adversarial examples using FGSM and I-FGSM. Then, we test the robustness of the second ERM model and the *mixup* model to these examples. The results of both settings are summarized in Table 3.

For the FGSM white box attack, the *mixup* model is 2.7 times more robust than the ERM model in terms of Top-1 error. For the FGSM black box attack, the *mixup* model is 1.25 times more robust than the ERM model in terms of Top-1 error. Also, while both *mixup* and ERM are not robust to white box I-FGSM attacks, *mixup* is about $40\%$ more robust than ERM in the black box I-FGSM setting. Overall, *mixup* produces neural networks that are significantly more robust than ERM against adversarial examples in white box and black settings without additional overhead compared to ERM.

| Dataset | ERM | *mixup* | | Dataset | ERM | *mixup* |
|---|---|---|---|---|---|---|
| Abalone | 74.0 | 73.6 | | Htru2 | 2.0 | 2.0 |
| Arcene | 57.6 | **48.0** | | Iris | 21.3 | **17.3** |
| Arrhythmia | 56.6 | **46.3** | | Phishing | 16.3 | 15.2 |

Table 4: ERM and *mixup* classification errors on the UCI datasets.

| ERM GAN | *mixup* GAN ($\alpha = 0.2$) |
|---|---|

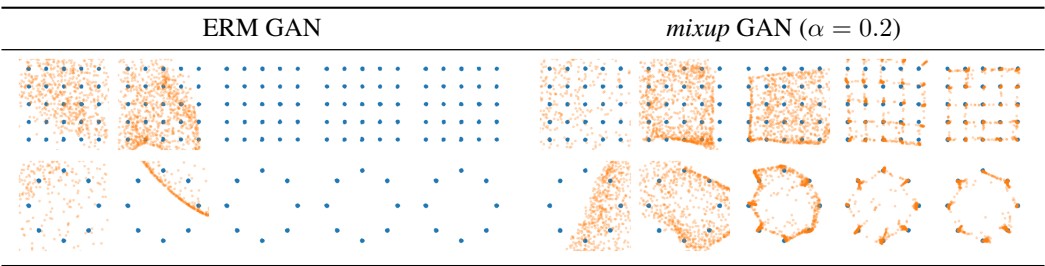

Figure 5: Effect of *mixup* on stabilizing GAN training at iterations 10, 100, 1000, 10000, and 20000.

### 3.6 TABULAR DATA

To further explore the performance of *mixup* on non-image data, we performed a series of experiments on six arbitrary classification problems drawn from the UCI dataset (Lichman, 2013). The neural networks in this section are fully-connected, and have two hidden layers of 128 ReLU units. The parameters of these neural networks are learned using Adam (Kingma & Ba, 2015) with default hyper-parameters, over 10 epochs of mini-batches of size 16. Table 4 shows that *mixup* improves the average test error on four out of the six considered datasets, and never underperforms ERM.

### 3.7 STABILIZATION OF GENERATIVE ADVERSARIAL NETWORKS (GANs)

Generative Adversarial Networks, also known as GANs (Goodfellow et al., 2014), are a powerful family of implicit generative models. In GANs, a generator and a discriminator compete against each other to model a distribution $P$. On the one hand, the generator $g$ competes to transform noise vectors $z \sim Q$ into fake samples $g(z)$ that resemble real samples $x \sim P$. On the other hand, the discriminator competes to distinguish between real samples $x$ and fake samples $g(z)$. Mathematically, training a GAN is equivalent to solving the optimization problem

$$\max_{g} \min_{d} \mathbb{E}_{x,z} \; \ell(d(x), 1) + \ell(d(g(z)), 0),$$

where $\ell$ is the binary cross entropy loss. Unfortunately, solving the previous min-max equation is a notoriously difficult optimization problem (Goodfellow, 2016), since the discriminator often provides the generator with vanishing gradients. We argue that *mixup* should stabilize GAN training because it acts as a regularizer on the gradients of the discriminator, akin to the binary classifier in Figure 1b. Then, the smoothness of the discriminator guarantees a stable source of gradient information to the generator. The *mixup* formulation of GANs is:

$$\max_{g} \min_{d} \mathbb{E}_{x,z,\lambda} \; \ell(d(\lambda x + (1 - \lambda)g(z)), \lambda).$$

Figure 5 illustrates the stabilizing effect of *mixup* the training of GAN (orange samples) when modeling two toy datasets (blue samples). The neural networks in these experiments are fully-connected and have three hidden layers of 512 ReLU units. The generator network accepts two-dimensional Gaussian noise vectors. The networks are trained for 20,000 mini-batches of size 128 using the Adam optimizer with default parameters, where the discriminator is trained for five iterations before every generator iteration. The training of *mixup* GANs seems promisingly robust to hyper-parameter and architectural choices.

| Method | Specification | Modified | | Weight decay | |
|---|---|---|---|---|---|
| | | Input | Target | $10^{-4}$ | $5 \times 10^{-4}$ |
| ERM | | ✗ | ✗ | 5.53 | 5.18 |
| *mixup* | AC + RP | ✓ | ✓ | **4.24** | 4.68 |
| | AC + KNN | ✓ | ✓ | 4.98 | 5.26 |
| mix labels and latent representations (AC + RP) | Layer 1 | ✓ | ✓ | 4.44 | **4.51** |
| | Layer 2 | ✓ | ✓ | 4.56 | 4.61 |
| | Layer 3 | ✓ | ✓ | 5.39 | 5.55 |
| | Layer 4 | ✓ | ✓ | 5.95 | 5.43 |
| | Layer 5 | ✓ | ✓ | 5.39 | 5.15 |
| mix inputs only | SC + KNN (Chawla et al., 2002) | ✓ | ✗ | 5.45 | 5.52 |
| | AC + KNN | ✓ | ✗ | 5.43 | 5.48 |
| | SC + RP | ✓ | ✗ | 5.23 | 5.55 |
| | AC + RP | ✓ | ✗ | 5.17 | 5.72 |
| label smoothing (Szegedy et al., 2016) | $\epsilon = 0.05$ | ✗ | ✓ | 5.25 | 5.02 |
| | $\epsilon = 0.1$ | ✗ | ✓ | 5.33 | 5.17 |
| | $\epsilon = 0.2$ | ✗ | ✓ | 5.34 | 5.06 |
| mix inputs + label smoothing (AC + RP) | $\epsilon = 0.05$ | ✓ | ✓ | 5.02 | 5.40 |
| | $\epsilon = 0.1$ | ✓ | ✓ | 5.08 | 5.09 |
| | $\epsilon = 0.2$ | ✓ | ✓ | 4.98 | 5.06 |
| | $\epsilon = 0.4$ | ✓ | ✓ | 5.25 | 5.39 |
| add Gaussian noise to inputs | $\sigma = 0.05$ | ✓ | ✗ | 5.53 | 5.04 |
| | $\sigma = 0.1$ | ✓ | ✗ | 6.41 | 5.86 |
| | $\sigma = 0.2$ | ✓ | ✗ | 7.16 | 7.24 |

Table 5: Results of the ablation studies on the CIFAR-10 dataset. Reported are the median test errors of the last 10 epochs. A tick (✓) means the component is different from standard ERM training, whereas a cross (✗) means it follows the standard training practice. AC: mix between all classes. SC: mix within the same class. RP: mix between random pairs. KNN: mix between k-nearest neighbors (k=200). Please refer to the text for details about the experiments and interpretations.

## 3.8    ABLATION STUDIES

*mixup* is a data augmentation method that consists of only two parts: random convex combination of raw inputs, and correspondingly, convex combination of one-hot label encodings. However, there are several design choices to make. For example, on how to augment the inputs, we could have chosen to interpolate the latent representations (i.e. feature maps) of a neural network, and we could have chosen to interpolate only between the nearest neighbors, or only between inputs of the same class. When the inputs to interpolate come from two different classes, we could have chosen to assign a single label to the synthetic input, for example using the label of the input that weights more in the convex combination. To compare *mixup* with these alternative possibilities, we run a set of ablation study experiments using the PreAct ResNet-18 architecture on the CIFAR-10 dataset.

Specifically, for each of the data augmentation methods, we test two weight decay settings ($10^{-4}$ which works well for *mixup*, and $5 \times 10^{-4}$ which works well for ERM). All the other settings and hyperparameters are the same as reported in Section 3.2.

To compare interpolating raw inputs with interpolating latent representations, we test on random convex combination of the learned representations before each residual block (denoted Layer 1-4) or before the uppermost "average pooling + fully connected" layer (denoted Layer 5). To compare mixing random pairs of inputs (RP) with mixing nearest neighbors (KNN), we first compute the 200 nearest neighbors for each training sample, either from the same class (SC) or from all the classes (AC). Then during training, for each sample in a minibatch, we replace the sample with a synthetic sample by convex combination with a random draw from its nearest neighbors. To compare mixing all the classes (AC) with mixing within the same class (SC), we convex combine a minibatch with a

random permutation of its sample index, where the permutation is done in a per-batch basis (AC) or a per-class basis (SC). To compare mixing inputs and labels with mixing inputs only, we either use a convex combination of the two one-hot encodings as the target, or select the one-hot encoding of the closer training sample as the target. For label smoothing, we follow Szegedy et al. (2016) and use $\frac{\epsilon}{10}$ as the target for incorrect classes, and $1 - \frac{9\epsilon}{10}$ as the target for the correct class. Adding Gaussian noise to inputs is used as another baseline. We report the median test errors of the last 10 epochs. Results are shown in Table 5.

From the ablation study experiments, we have the following observations. First, *mixup* is the best data augmentation method we test, and is significantly better than the second best method (mix input + label smoothing). Second, the effect of regularization can be seen by comparing the test error with a small weight decay ($10^{-4}$) with a large one ($5 \times 10^{-4}$). For example, for ERM a large weight decay works better, whereas for *mixup* a small weight decay is preferred, confirming its regularization effects. We also see an increasing advantage of large weight decay when interpolating in higher layers of latent representations, indicating decreasing strength of regularization. Among all the input interpolation methods, mixing random pairs from all classes (AC + RP) has the strongest regularization effect. Label smoothing and adding Gaussian noise have a relatively small regularization effect. Finally, we note that the SMOTE algorithm (Chawla et al., 2002) does not lead to a noticeable gain in performance.

## 4 RELATED WORK

Data augmentation lies at the heart of all successful applications of deep learning, ranging from image classification (Krizhevsky et al., 2012) to speech recognition (Graves et al., 2013; Amodei et al., 2016). In all cases, substantial domain knowledge is leveraged to design suitable data transformations leading to improved generalization. In image classification, for example, one routinely uses rotation, translation, cropping, resizing, flipping (Lecun et al., 2001; Simonyan & Zisserman, 2015), and random erasing (Zhong et al., 2017) to enforce visually plausible invariances in the model through the training data. Similarly, in speech recognition, noise injection is a prevalent practice to improve the robustness and accuracy of the trained models (Amodei et al., 2016).

More related to *mixup*, Chawla et al. (2002) propose to augment the rare class in an imbalanced dataset by interpolating the nearest neighbors; DeVries & Taylor (2017) show that interpolation and extrapolation the nearest neighbors of the same class in feature space can improve generalization. However, their proposals only operate among the nearest neighbors within a certain class at the input / feature level, and hence does not account for changes in the corresponding labels. Recent approaches have also proposed to regularize the output distribution of a neural network by label smoothing (Szegedy et al., 2016), or penalizing high-confidence softmax distributions (Pereyra et al., 2017). These methods bear similarities with *mixup* in the sense that supervision depends on multiple smooth labels, rather than on single hard labels as in traditional ERM. However, the label smoothing in these works is applied or regularized independently from the associated feature values.

*mixup* enjoys several desirable aspects of previous data augmentation and regularization schemes without suffering from their drawbacks. Like the method of DeVries & Taylor (2017), it does not require significant domain knowledge. Like label smoothing, the supervision of every example is not overly dominated by the ground-truth label. Unlike both of these approaches, the *mixup* transformation establishes a linear relationship between data augmentation and the supervision signal. We believe that this leads to a strong regularizer that improves generalization as demonstrated by our experiments. The linearity constraint, through its effect on the derivatives of the function approximated, also relates *mixup* to other methods such as Sobolev training of neural networks (Czarnecki et al., 2017) or WGAN-GP (Gulrajani et al., 2017).

## 5 DISCUSSION

We have proposed *mixup*, a data-agnostic and straightforward data augmentation principle. We have shown that *mixup* is a form of vicinal risk minimization, which trains on virtual examples constructed as the linear interpolation of two random examples from the training set and their labels. Incorporating *mixup* into existing training pipelines reduces to a few lines of code, and introduces little or no computational overhead. Throughout an extensive evaluation, we have shown that *mixup*

improves the generalization error of state-of-the-art models on ImageNet, CIFAR, speech, and tabular datasets. Furthermore, *mixup* helps to combat memorization of corrupt labels, sensitivity to adversarial examples, and instability in adversarial training.

In our experiments, the following trend is consistent: with increasingly large $\alpha$, the training error on real data increases, while the generalization gap decreases. This sustains our hypothesis that *mixup* implicitly controls model complexity. However, we do not yet have a good theory for understanding the 'sweet spot' of this bias-variance trade-off. For example, in CIFAR-10 classification we can get very low training error on real data even when $\alpha \to \infty$ (i.e., training *only* on averages of pairs of real examples), whereas in ImageNet classification, the training error on real data increases significantly with $\alpha \to \infty$. Based on our ImageNet and Google commands experiments with different model architectures, we conjecture that increasing the model capacity would make training error less sensitive to large $\alpha$, hence giving *mixup* a more significant advantage.

*mixup* also opens up several possibilities for further exploration. First, is it possible to make similar ideas work on other types of supervised learning problems, such as regression and structured prediction? While generalizing *mixup* to regression problems is straightforward, its application to structured prediction problems such as image segmentation remains less obvious. Second, can similar methods prove helpful beyond supervised learning? The interpolation principle seems like a reasonable inductive bias which might also help in unsupervised, semi-supervised, and reinforcement learning. Can we extend *mixup* to feature-label extrapolation to guarantee a robust model behavior far away from the training data? Although our discussion of these directions is still speculative, we are excited about the possibilities *mixup* opens up, and hope that our observations will prove useful for future development.

## ACKNOWLEDGEMENTS

We would like to thank Priya Goyal, Yossi Adi and the PyTorch team. We also thank the Anonymous Review 2 for proposing the *mixup* + dropout experiments.

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
