# OpenReview forum: "mixup: Beyond Empirical Risk Minimization"
_ICLR.cc/2018/Conference — Accept (Poster)_

### Official Review · AnonReviewer2 · 2017-11-22
**Baseline should be mixing features only**

**Rating:** 6
**Confidence:** 4

**Review:**

This paper studies an approach of data augmentation where a convex combination of multiple samples is used as a new sample.  While the use of such convex combination (mixing features) is not new, this paper proposes to use a convex combination of corresponding labels as the label of the new sample (mixing labels).  The authors motivate the proposed approach in the context of vicinal risk minimization, but the proposed approach is not well supported by theory.  Experimental results suggest that the proposed approach significantly outperforms the baseline of using only the standard data augmentation studied in Goyal et al. (2017).

While the idea of mixing not only features but also labels is new and interesting, its advantage over the existing approach of mixing only features is not shown.  The authors mention "interpolating only between inputs with equal label did not lead to the performance gains of mixup," but this is not shown in the experiments.  The authors cite recent work by DeVries & Taylor (2017) and Pereyra et al. (2017), but the technique of combining multiple samples for data augmentation have been a popular approach.  See for example a well cited paper by Chawla et al. (2002).  The baseline should thus be mixing only features, and this should be compared against the proposed approach of mixing both features and labels.

N. V. Chawla et al., SMOTE: Synthetic Minority Over-sampling Technique, JAIR 16: 321-357 (2002).

Minor comments:

Figure 1(b): How should I read this figure?  For example, what does the color represent?

Table 1: What is an epoch for mixup?  How does the per epoch complexity of mixup copare against that of ERM?

Table 2: The test error seems to be quite sensitive to the number of epochs.  Why not use validation to determine when to stop training?

Table 2: What is the performance of mixup + dropout?

===

I appreciate the thorough revision.  The empirical advantages over baselines including SMOTE and others are now well demonstrated in the experimental results.  It is also good to see that mixup is complementary to dropout, and the combined method works even better than either.

I understand and appreciate the authors' argument as to why mixup should work, but it is not sufficiently convincing to me why a convex combination in Euclidean space should produce good data distribution.  Convex combination certainly changes the manifold.  However, the lack of sufficient theoretical justification is now well complemented by extensive experiments, and it will motivate more theoretical work.

---

> ### Author Response · Authors · 2018-01-04
> **clarification of differences from previous work; design rationale and ablation studies**
>
> We thank the Anonymous Reviewer 2 for comments and feedback.
>
> A major concern of Reviewer 2 is that the paper should include "mixing features" as a baseline.
>
> First of all, we thank Reviewer 2 for raising this point and referring to the SMOTE paper. In the latest revised version (now available for download), we have included a new ablation study section which thoroughly compares mixup against related data augmentation ideas and gives further support to mixup's advantage.
>
> Moreover, we would like to clarify that mixup and previous work have other important differences. Specifically, the SMOTE algorithm only makes convex combinations of the raw inputs between *nearest neighbors of the same class*. The recent work by DeVries & Taylor (2017) also follows the same-class nearest neighbor idea of SMOTE, albeit in the feature space (e.g. the embedding space of an autoencoder). This is in sharp contrast to our proposed mixing strategy, which makes convex combination of randomly drawn raw inputs pairs from the training set. In the revised submission, we highlight these differences and demonstrate that each of them is essential for achieving better performance.
>
> Conceptually, a desideratum of data augmentation is that the augmented data should be as diverse as possible while covering the space of data distribution. In the case where the data distribution is a low dimensional manifold (such as the Swiss Roll Dataset), typically the number of data n and the dimensionality of the input space d satisfy d << log(n), it suffices to augment the training set by interpolating the nearest neighbors. However, if on the other hand d >> log(n), as is the case in typical vision and speech classification tasks, nearest neighbors provide insufficient information to recover the geometry of the data distribution, and training samples other than nearest neighbors can provide a lot of additional geometric information of the data distribution. Therefore, compared with interpolating nearest neighbors (of either the same class, or the entire training set), interpolating random pairs of training data provides a better coverage of the data distribution. Empirically, in our new ablation studies, we find that mixing nearest neighbors provides little (if any) improvement over ERM.
>
> As we have justified the interpolation of random training data pairs from potentially different classes, it remains to decide which loss function to use if the synthetic data is a convex combination of samples from two classes. One choice is to assign a single label to the synthetic sample, presumably using the label of the closer sample from the two inputs used to generate the synthetic sample. However, this choice creates an abrupt change of target around a 50-50 mix, while not being able to distinguish a 55-45 mix from a 95-5 mix. The natural solution to these two problems is to also mix the labels using the same weights as the input mix, which makes sure a 55-45 mix of inputs is more similar to a 45-55 mix, rather than a 95-5 one. Empirically, in our new ablation studies, we find that "only mixing the inputs" provides some regularization effects (so that the performance of using smaller weight decay improves), but very limited performance gain over ERM.
>
> Please also refer to the reply to Reviewer 3 for more theoretical justifications.
>
> Regarding the "minor comments":
>
> Figure 1(b): - Green: Class 0, Orange: Class 1, Blue shading indicates p(y=1). It is now clarified in the revised version.
>
> Table 1: One epoch of mixup training is the same as one epoch of normal training, with the same number of minibatches and the same minibatch size. The only change in the training loop is the input mixing step and the label mixing step. Therefore, the computational complexity and actually training time remain (almost) the same.
>
> Table 2: For these experiments, the dataset contains a large portion of corrupt labels. In this case, *without proper regularization*, the test error is indeed quite sensitive to the number of epochs, and one can use cross-validation to determine when to stop. However, even if we only consider the best test errors achieved during the training process, mixup still has a significant advantage over dropout, and dropout has a significant advantage over ERM.
>
> Table 2: What is the performance of mixup + dropout?
> This is a very good question. We conduct additional experiments combining mixup and dropout, both with medium regularization strength. We observe that the best parameter setting of the combined method is comparable with mixup in terms of the best test error during the training process, but outperforms mixup in terms of the test error at the last epoch. This suggests that mixup combined with dropout is even more resistant to corrupt labels. The updated results are available now. We thank Reviewer 2 for raising this interesting question, and will include proper acknowledgement in the final version.

---

### Official Review · AnonReviewer1 · 2017-11-24
**Interesting strategy**

**Rating:** 7
**Confidence:** 4

**Review:**

I enjoyed reading this well-written and easy-to-follow paper. The paper builds on the rather old idea of minimizing the empirical vicinal risk (Chapelle et al., 2000) instead of the empirical risk. The authors' contribution is to provide a particular instance of vicinity distribution, which amounts to linear interpolation between samples. This idea of linear interpolation on the training sample to generate additional (adversarial, in the words of the authors) data is definitely appealing to prevent overfitting and improve generalization performance at a mild computational cost (note that this comment does not just apply to deep learning). This notion is definitely of interest to machine learning, and to the ICLR community in particular. I have several comments and remarks on the concept of mixup, listed below in no particular order. My overall opinion on the paper is positive and I stand for acceptance, provided the authors answer the points below. I would especially be interested in discussing those with the authors.

1 - While data augmentation literature is well acknowledged in the paper, I would also like to see a comment on domain adaptation, which is a very closely related topic and of particular interest to the ICLR community.

2 - Paragraph after Eq. (1), starting with "Learning" and ending with "(Szegedy et al., 2014)": I am not so familiar with the term memorization, is this just a fancy way of talking about overfitting? If so, you might want to rephrase this paragraph with terms more used in the machine learning community. When you write "one trivial way to minimize [the empirical risk] is to memorize the training data", do you mean output a predictor which only delivers predictions on $X_i$, equal to $Y_i$? If so, this is again not specific to deep learning and I feel this should be a bit more discussed.

3 - I have not found in the paper a clear heuristics about how pairs of training samples should be picked to create interpolations. Picking at random is the simplest however I feel that a proximity measure on the space $\mathcal{X}$ on which samples live would come in handy. For example, sampling with a probability decreasing as the Euclidean distance seems a natural idea. In any case, I strongly feel this discussion is missing in the paper.

4 - On a related note, I would like to see a discussion on how many "adversarial" examples should be used. Since the computational overhead cost of computing one new sample is reasonable (sampling from a Beta distribution + one addition), I wonder why $m$ is not taken very large, yielding more accurate estimates of the empirical risk. A related question: under what conditions does the vicinal risk converge (in expectation for example) to the empirical risk? I think some comments would be nice.

5 - I am intrigued by the last paragraph of Section 5. What do the authors exactly have in mind when they suggest that mixup could be generalized to regression problems? As far as I understood the paper, since $\tilde{y}$ is defined as a linear interpolation between $y_i$ and $y_j$, this formulation only works for continuous $y$s, like in regression. This formulation is not straightforwardly transposable to classification for example. I therefore am quite confused about the fact that the authors present experiments on classification tasks, with a method that writes for regression.

6 - Writing linear interpolations to generate new data points implicitly makes the assumption that the input and output spaces ($\mathcal{X}$ and $\mathcal{Y}$) are convex. I have no clear intuition wether this is a limitation of the authors' proposed method but I strongly feel this should be carefully addressed by a comment in Section 2.

---

> ### Author Response · Authors · 2018-01-05
> **answers to your interesting questions**
>
> We thank the Anonymous Reviewer 1 for interesting comments and feedback.
>
> 1. Domain adaptation is indeed a related problem as one can consider a model trained with Vicinal Risk Minimization will be more robust to small drift in the input distribution. The experience on adversarial examples tends to validate this hypothesis. Indeed, the distance between the original distribution and that of the adversarial examples is typically small. A full study of the suitability of mixup to the broader domain adaptation problems is however beyond the scope of this paper.
>
> We now include a brief discussion about domain adaptation. Inspired by this question, we brainstormed about the possibility of using mixup for domain adaptation in two ways.
>
> (1) Assume a large source-domain dataset D_s = { (xs_1, ys_1), ..., (xs_N, ys_N) }, and a small target-domain dataset D_t = { (xt_1, yt_1), ..., (xt_n, yt_n) }. We are interested in learning D_t with auxiliary knowledge from D_s. Using mixup we could do so by training our classifier on the synthetic pairs:
>
> x = a * xt_i + (1 - a) * xs_j,
> y = a * yt_i + (1 - a) * xs_j,
>
> For random pairs of indices i \in { 1, …, n }, j \in { 1, …, N }, and a particular distribution for the mixing coefficient a. If a is concentrated around one, this process recovers ERM training on the target domain. Otherwise, mixup produces synthetic examples just outside the target domain, by interpolating into the source domain.
>
> (2) Alternatively, mixup can simply be used as a within-domain data augmentation method in existing domain adaptation algorithms. For example, some recent work (e.g. https://arxiv.org/abs/1702.05464) train a network to have similar embedding distributions for both the source and the target domains, using shared weights, moment matching or discriminator loss. With mixup, we can use the same mixing weights lambda (or hyperparameter alpha) for both the source and the target domain samples, and require the learned embeddings to be similar. This forces the model to match the embedding distributions of two domains at a broader region in the embedding space, potentially improving the transfer performance.
>
> 2. Correct, by memorization we mean perfect overfitting, as discussed in (Zhang et al., 2017). We will clarify this issue, as well as mentioning that this is a pathology more general than deep learning.
>
> 3. At the core of mixup lies its simplicity: pairs of training examples are chosen at random. We have considered other possibilities based on nearest neighbours, but random pairing was much simpler (it does not require to specify a norm in X), and produced better results. We now dedicate a paragraph in the manuscript to describe our choice, as well as proposing the ones by the reviewer for future work.
>
> 4. Mixup does not involve computing adversarial examples. Instead, mixup constructs synthetic examples by interpolating random pairs of points from the training set. New synthetic examples are constructed on-the-fly at a negligible cost for each training iteration. Note that *adversarial* examples are only constructed in our experiment to verify the robustness of mixup networks to adversarial attacks. Adversarial examples are never constructed during training. They are only constructed in that particular experiment at test time, to verify the robustness of each network.
>
> For some examples of VRM converging to ERM, we suggest the original paper of Chapelle et al. An example discussed in that paper is the equivalence of adding Gaussian perturbation to inputs and ERM with L2 regularization.
>
> 5. Mathematically, mixup works for both classification and regression problems: for classification, all our experiments mix labels when parameterized as one-hot continuous vectors. For example: 0.3 * (0, 1, 0) + 0.7 * (1, 0, 0) = (0.7, 0.3, 0). However, we haven't done any experiments on regression problems, and therefore we are interested to see i) if regression performance improves with mixup, and ii) how the regression curves are regularized by mixup.
>
> 6. We do not make a convexity assumption, in the sense that linear combination of samples fall outside the set of natural images, and therefore we are forcing the classifier to give reasonable (mixup) labels "outside the convex set". Overall, linear interpolation is not a limitation, but a simple and powerful way to inform the classifier about how the label changes in the neighbourhood of an image (“this is one direction along which a ‘cat’ becomes closer to a ‘dog’”, etc). In the case where the input space is more structural (e.g. the space of graphs), a convex combination of the raw inputs may not be valid. However, we can always make the convex combination in the embedding space, which is supposed to be a vector space.

---

> > ### Public Comment · (anonymous) · 2018-01-18
> > **How does mixing of labels work?**
> >
> > Dear authors,
> >
> > I came across your paper when I was doing literature review for my own project. I really enjoyed it as it's well written, easy to follow and showed great empirical performance. I would love to try this in my own project, however, I got confused by the label mixing part for classification task.
> >
> > Following up on your explanation 5), you obtained a smoothed label as (0.7, 0.3, 0). I also found in your paper you claimed that "To compare mixing inputs and labels with mixing inputs only, we either use a convex combination of the two one-hot encodings as the target, or select the one-hot encoding of the closer training sample as the target." The below points are based on my speculation:
> >
> > If you used this float value directly, can you elaborate more on what type of loss function did you apply to train your network?
> >
> > If you used one-hot encoding that is closer to the training target, if I am understanding correctly, (0.7, 0.3, 0) is selected as (1, 0, 0). Then all the mix-up labels would only depend on the values of alpha: y = alpha y_1 + (1-alpha) y_2, if alpha > 0.5, then y = y1, and if alpha < 0.5, y = y2. The mix-up procedure would produce a training sample like ( alpha x_1 + (1-alpha) x_2, y_1), which feels like adding perturbations to the input.
> >
> > I hope I am understanding your paper correctly and would love to hear your explanation. Thanks so much!

---

> > > ### Author Response · Authors · 2018-01-19
> > > **cross-entropy loss is a function of two discrete distributions**
> > >
> > > Thanks for your interests in our work!
> > >
> > > We could use the floating point value directly with the cross-entropy loss. Note that the cross-entropy loss function (https://en.wikipedia.org/wiki/Cross_entropy) can be written as:
> > > l(p, q) = \sum_i p_i * log(q_i) = p^T log(q)           (1)
> > > where i is the category index (e.g. class label index) and p, q are two discrete distributions.
> > >
> > > - In typical ERM training, p is the one-hot encoding of the target, and q is the predicted softmax probabilities.
> > > - If we convexly combine two one-hot target vectors y1 and y2, i.e. p = lam * y1 + (1 - lam) * y2, then since the resulting p represents a discrete distribution, (1) still works without modification. (For example, in Tensorflow you can feed p into `tf.nn.softmax_cross_entropy_with_logits`, as suggested in https://github.com/tensorflow/tensorflow/blob/abf3c6d745c34d303985f210bf9e92cac99ba744/tensorflow/python/ops/nn_ops.py#L1713 ; in PyTorch you may need to implement (1) by writing a customized loss function.)
> > > - Alternatively, note that l(p, q) is linear in p, which means in the above case, l(p, q) = l(lam * y1 + (1 - lam) * y2, q) = lam * l(y1, q) + (1 - lam) * l(y2, q). Hence you can simply use the cross-entropy loss function in your favorite ML framework to compute l(y1, q) and l(y2, q), as you do for ERM, and then use lam to convexly combine them to get l(p, q). This is what we use in our implementation.
> > >
> > > Finally, note that label smoothing (https://www.cv-foundation.org/openaccess/content_cvpr_2016/papers/Szegedy_Rethinking_the_Inception_CVPR_2016_paper.pdf) can also be implemented using (1) by writing a customized loss function.
> > >
> > > Hope the above explanation is clear. Thanks!

---

### Official Review · AnonReviewer3 · 2017-12-02
**Good results, but important baselines missing and no backing theory**

**Rating:** 6
**Confidence:** 4

**Review:**

Theoretical contributions: None. Moreover, there is no clear theoretical explanation for why this approach ought to work. The authors cite (Chapelle et al., 2000) and actually most of the equations are taken from there, but the authors do not justify why the proposed distribution is a good approximation for the true p(x, y).

Practical contributions: The paper introduces a new technique for training DNNs by forming a convex combination between two training data instances, as well as changing the associated label to the corresponding convex combination of the original 2 labels.

Experimental results. The authors show mixup provides improvement over baselines in the following settings:
  * Image Classification on Imagenet. CIFAR-10 and CIFAR-100, across architectures.
  * Speech data
  * Memorization of corrupted labels
  * Adversarial robustness (white box and black box attacks)
  * GANs (though quite a limited example, it is hard to generalize from this setting to the standard problems that GANs are used for).
  * Tabular data.

Reproducibility: The provided website to access the source code is currently not loading. However, experiment hyperparameters are meticulously recorded in the paper.

Key selling points:
  * Good results across the board.
  * Easy to implement.
  * Not computationally expensive.

What is missing:
  * Convincing theoretical arguments for why combining data and labels this way is a good approach. Convex combinations of natural images does not result in natural images.
 * Baseline in which the labels are not mixed, in order to ensure that the gains are not coming from the data augmentation only. Combining the proposed data augmentation with label smoothing should be another baseline.
  * A thorough discussion on mixing in feature space, as well as a baseline which mizes in feature space.
  * A concrete strategy for obtaining good results using the proposed method. For example, for speech data the authors say that “For mixup, we use a warm-up period of five epochs where we train the network on original training examples, since we find it speeds up initial convergence.“ Would be good to see how this affects results and convergence speed. Apart from having to tune the lambda hyperparameter, one might also have to tune when to start mixup.
  * Figure 2 seems like a test made to work for this method and does not add much to the paper. Yes, if one trains on convex combination between data, one expects the model to do better in that regime.
  * Label smoothing baseline to put numbers into perspective, for example in Figure 4.

---

> ### Author Response · Authors · 2018-01-05
> **explanation of the design choices; ablation studies**
>
> We thank the Anonymous Reviewer 3 for comments and feedback.
>
> The following exposition concerns theoretical arguments for the proposed approach, as well as empirical comparisons with mixing only the inputs, mixing in feature space, label smoothing (with and without mixing the inputs). We include all these baselines in an ablation study section in the latest revision (now available for download), giving further support to mixup's advantage. We now explain the theoretical motivations:
>
> One desideratum of data augmentation is that the augmented data should match the statistics of the training set. This is because augmented data that mismatch the statistics of the training set are not likely to match the statistics of the test set, leading to less effective augmentation. From the perspective of training, using augmented data that deviate too much from the training set statistics will also cause *high training error on the original training set*, and in turn hurt generalization. In the following, we argue that input space interpolation better matches the statistics of the training set.
>
> Classifiers are nothing but functions of the input space. Therefore, similarity between the statistics of the augmented data and those of the data distribution in input space is more important than perceptual quality or semantic interpretability. Perceptual similarity does not correlate well with metric distance in the input space. For example, a blurred image usually looks similar to its original sharp version, while in input space they may have a large (L2) distance. On the other hand, by directly interpolating in the input space, our method is bound to not lose or significantly alter the statistical information in the data distribution. Therefore, despite not looking real, the mixup images can be better synthetic data for training classifiers than images from latent space interpolations. Empirically, in our ablation studies, we see a gradual degradation of accuracy when interpolating in higher layers of representation.
>
> Another desideratum is that the augmented data should be as diverse as possible to cover the space spanned by the training set. In the case where the data distribution is a low dimensional manifold (such as the Swiss Roll Dataset), typically the number of data n and the dimensionality of the input space d satisfy d << log(n), it suffices to augment the training set by interpolating the nearest neighbors. However, if on the other hand d >> log(n), as is the case in typical vision and speech classification tasks, nearest neighbors provide insufficient information for recovering the geometry of the data distribution, and training samples other than nearest neighbors can provide a lot of additional geometric information of the data distribution. Therefore, compared with interpolating nearest neighbors, interpolating random pairs of training data provides a better coverage of the data distribution. Empirically, in our ablation studies, we find that mixing nearest neighbors provides little (if any) improvement over ERM.
>
> As we have justified the interpolation of training data from potentially different classes, it remains to decide which synthetic label to use if the synthetic input is a convex combination of samples from two classes. One choice is to assign a single label to the synthetic sample, presumably using the label of the closer sample from the two inputs used to generate the synthetic sample. However, this choice creates an abrupt change of target around a 50-50 mix, while not being able to distinguish a 55-45 mix from a 95-5 mix. The natural solution to these two problems is to also mix the labels using the same weights, which makes sure a 55-45 mix of inputs is more similar to a 45-55 mix, rather than a 95-5 one. Empirically, in our ablation studies, we find that only mixing the inputs provides some regularization effects, but very limited performance gain over ERM.
>
> One might also consider replacing the label interpolation in mixup with label smoothing or similar target space regularizers (e.g. Pereyra et al., 2017). However, label smoothing and similar work are not designed for between-class input interpolation in that it assigns a small but fixed amount of probability to every class that is not the correct class. Therefore, this loss also has the two problems mentioned above. Empirically, in our ablation studies, we find that adding label smoothing to ERM or replacing label interpolation with label smoothing in mixup provide only limited performance gain over ERM.
>
> Other comments:
>
> "warm-up": we believe using SGD with momentum and the standard learning rate schedule (i.e. reducing the learning rate when training error plateaus) is sufficient for good performance.
>
> "Figure 2": it shows that the ERM model has improper behaviors not only in adversarial directions but also between training points, which to the best of our knowledge is not commonly known. It also provides a direct motivation for mixup.

---

### Public Comment · (anonymous) · 2017-11-06
**Related submission**

I am an author of Paper275.
I found my submission seems tightly related to this work. So I put a link to my submission for your information,
Data Augmentation by Pairing Samples for Images Classification
https://openreview.net/forum?id=SJn0sLgRb&noteId=SJn0sLgRb

---

### Author Response · Authors · 2018-01-05
**changes in the revision (Jan 5, 2018)**

As per the request of reviewers, we now include a new "ablation studies" section in the (Jan 5, 2018) revision (available for download). The new experiments compare mixup with mixing only the inputs (of either random pairs or nearest neighbors, with the pair of samples coming from either the same class or any classes. This includes a previous method called "SMOTE"), mixing in feature space, label smoothing (with and without mixing the inputs) and adding Gaussian noise to inputs. The results confirm the advantage of mixup over other design choices.

For theoretical justifications of the design choices of mixup, please refer to our replies to the reviewers, in particular Reviewer 2 and 3. Thanks for your interests!

---

### Decision · Program_Chairs · 2018-01-29
**ICLR 2018 Conference Acceptance Decision**

**Decision:**

Accept (Poster)

**Comment:**

The paper presents a simple but surprisingly effective data augmentation technique which is thoroughly evaluated on a variety of classification tasks, leading to improvement over state-of-the-art baselines. The paper is somewhat lacking a theoretical justification beyond intuitions, but extensive evaluation makes up for that.